# FROM SINGLE TO DUAL REFERENCE: REINFORCEMENT-ALIGNED MULTI-IMAGE INSTRUCTION-GUIDED EDITING

## ABSTRACT

**Abstract**. Most instruction-guided image editing models assume a single reference image. However, many real-world tasks—such as combining people into a group portrait, integrating a subject into a scene, or transferring clothing between individuals—require reasoning across multiple inputs. Current approaches either fail outright or rely on ad-hoc heuristics to merge results. In this work, we present the systematic study of dual-image instruction-guided editing. To support this, we construct a synthesized dataset of dual-image instructions spanning five representative categories: animal–scene composition, person–scene insertion, group portraits, style transfer, and clothing replacement. Building on the open-source single-image reference editing model, we introduce a dual positional embedding scheme with LoRA fine-tuning that enables efficient multi-reference fusion without catastrophic forgetting. Furthermore, we apply reinforcement alignment with Diffusion Denoising Policy Optimization (DDPO), using vision language model as a reward model to better align generations with editing instructions. Despite being trained on relatively small-scale data, our method achieves strong qualitative and quantitative improvements, surpassing existing open source baselines in multi-reference editing. We release the dataset as the first benchmark for dual-image editing and provide an online demo to encourage further research in this direction (*here*).

## 1 INTRODUCTION

Instruction-guided image editing has rapidly progressed with diffusion-based approaches.(Brooks et al., 2023)(Zhao et al., 2024)(Zhang et al., 2024)(Shi et al., 2024)(Zhang et al., 2023) These models allow users to modify an input image according to natural-language instructions, enabling tasks such as object insertion, style transfer, and content removal.

In parallel, several closed-source commercial systems have demonstrated impressive editing capabilities, including GPT-4o, Gemini-Image-Flash 2.5, SeedEdit, and Flux Kontext.(OpenAI, 2024)(Google, 2025)(Wang et al., 2025a)(Labs et al., 2025) Among these, GPT-4o shows some ability to handle multiple image references. However, aside from this exception, most closed models do not explicitly support multi-image reference editing. On the open-source side, a number of strong editing frameworks have emerged: Early instruction-driven diffusion editors such as SDEdit(Meng et al., 2022) , InstructPix2Pix(Brooks et al., 2023) , and DiffEdit(Couairon et al., 2022). More recent developments including Qwen-Image-Edit(Wu et al., 2025a), Step-Edit(Liu et al., 2025) and the open-source release of Flux Kontext-dev from Black Forest Labs.

These open-source systems provide a powerful base for research into multi-image extensions. Yet, none currently offer direct support for universal dual- or multi-image reference editing, despite its importance in realistic scenarios (e.g., merging individuals into a single scene, transferring clothing styles across subjects).

In this paper, we take a step toward filling this gap. Specifically, we contribute:

- Task design and synthetic benchmark: We design a set of dual-image editing tasks — including animal–scene composition, person–scene insertion, group portraits, style transfer, and clothing replacement. Using publicly available open datasets and GPT-based synthe-

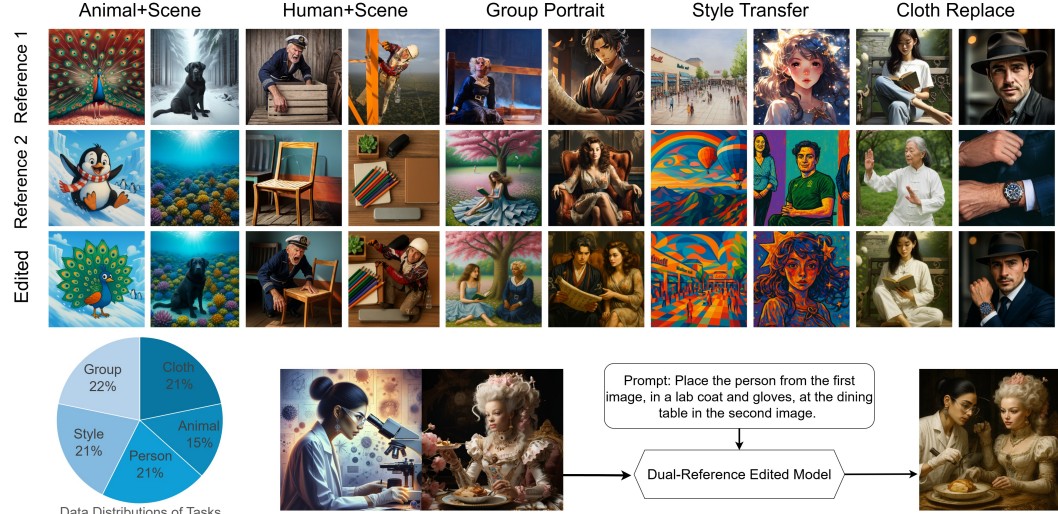

Figure 1: Overview of dataset and tasks. (a) Five tasks: animal–scene insertion, person–scene insertion, group portraits, style transfer, and clothing replacement. (b) Dataset distribution. (c) Task: generate an edited image from instructions using two reference images.

sis, we construct high-quality paired instructions and editing structures, forming the first benchmark for dual-image editing.

- Edit model extension: We extend the single-image reference edit architecture by redesigning its positional embedding scheme to allow lossless multi-image fusion. Through efficient LoRA fine-tuning, the model gains dual-image editing ability without catastrophic forgetting of its single-image capabilities.

- Reinforcement alignment with vision language model(VLM): To further improve instruction-following, we employ reinforcement learning with VLM as a vision–language evaluator. This reinforcement alignment significantly enhances edit fidelity and responsiveness.
  Together, these advances enable open-source model to achieve robust multi-image editing, demonstrating that multi-reference fusion is feasible even with relatively modest training data. Our dataset also provides the first standardized benchmark for evaluating future models in this setting.

## 2 RELATED WORK

### 2.1 INSTRUCTION-GUIDED IMAGE EDITING

Instruction-based diffusion models have enabled flexible image editing through natural-language commands. Early approaches such as SDEdit(Meng et al., 2022) used stochastic differential equations to gradually modify images while preserving structure. DiffEdit(Couairon et al., 2022) proposed semantic mask discovery to localize edits. InstructPix2Pix(Brooks et al., 2023) leveraged paired "before–after" image–instruction data to train models capable of following natural editing commands. More recently, MagicBrush(Zhang et al., 2024) introduced a large-scale dataset for free-form editing, further improving the diversity of instruction-following behavior.

While effective, these methods are restricted to single-image inputs. They excel at style transfer, local modifications, or content removal, but cannot incorporate additional references — a limitation in scenarios such as merging multiple people into one portrait or inserting objects consistently into scenes.

## 2.2 LARGER-SCALE OPEN-SOURCE EDITING FRAMEWORKS

More recently, larger-scale open instruction-following models have appeared. Qwen-Image-Edit provides a strong vision–language editing pipeline trained on large-scale synthetic data, supporting diverse text-to-image transformations. Meanwhile, Flux Kontext-Dev represents one of the strongest publicly available editing backbones, with high-quality results and flexible editing APIs. These models provide excellent base architectures for exploring multi-image extensions. However, none natively support multi-reference editing, nor is there a standardized dataset for evaluating such capabilities.

## 2.3 MULTI-IMAGE REFERENCE EDITING AND DATASETS

Recent studies extend personalization beyond single references toward multi-subject or multi-reference editing. MUSAR(Guo et al., 2025) introduces attention routing for multi-subject customization, RealCustom++(Mao et al., 2024) represents references as text tokens for efficient customization, MS-Diffusion(Wang et al., 2025b) incorporates layout guidance for multi-subject generation, and Less-to-More Generalization(UNO)(Wu et al., 2025b) explores in-context controllability. While promising, these works often focus on narrow tasks such as clothing replacement or merging two subjects, and lack the ability to handle complex instruction-driven multi-image editing.

On the data side, several open-source datasets have fueled single-image editing, including HQ-Edit(Hui et al., 2024), OmniEdit(Wei et al., 2025), UltraEdit, and ByteMorph(Chang et al., 2025). These resources support fine-grained or motion-based edits, but remain limited to single-reference scenarios. MUSAR is one of the few datasets addressing multi-subject customization, yet it does not target general instruction-based dual-image editing. In summary, while single-image editing has strong dataset and model support, benchmarks for multi-image reference editing remain absent. Our work fills this gap by introducing a synthetic dual-image dataset and method tailored to instruction-driven fusion tasks.

## 2.4 REINFORCEMENT LEARNING FOR EDITING ALIGNMENT

Reinforcement learning has been shown to improve alignment in generative models. Reinforcement Learning from Human Feedback (RLHF)(Christiano et al., 2023) is widely adopted in large language models. For diffusion models, Diffusion Denoising Policy Optimization (DDPO)(Black et al., 2024) provides a framework for optimizing policies with non-differentiable reward signals. Our approach adopts DDPO with GPT-5(OpenAI, 2025) as a multimodal evaluator, aligning dual-image editing outputs with natural instructions.

## 3 METHOD

Our method aims to extend open-source instruction-guided editing models toward dual-image reference editing. The pipeline consists of three main components: (1) construction of a synthetic dual-image editing dataset, (2) architectural modifications to the open source edit model for lossless multi-image fusion, and (3) reinforcement alignment using DDPO with GPT-5 as a vision–language reward model.

### 3.1 SYNTHETIC DATASET FOR DUAL-IMAGE EDITING

A major challenge for dual-image editing is the absence of suitable training data. While several high-quality datasets exist for single-image instruction editing—such as HQ-Edit, OmniEdit, UltraEdit, and ByteMorph—they are all restricted to single-reference scenarios. Efforts like MUSAR explore multi-subject customization, but focus on narrow personalization tasks rather than general instruction-driven multi-image editing. To bridge this gap, we construct a synthetic dual-image benchmark, using GPT-based instruction generation combined with open-source image datasets, enabling systematic study of multi-reference editing.

**Task design**. We consider five representative dual-image editing tasks as visualized in Figure 1. The first is *animal–scene composition*, where an animal must be inserted into a new scene while preserving scale and lighting. Second, *person–scene insertion* requires placing an individual into a

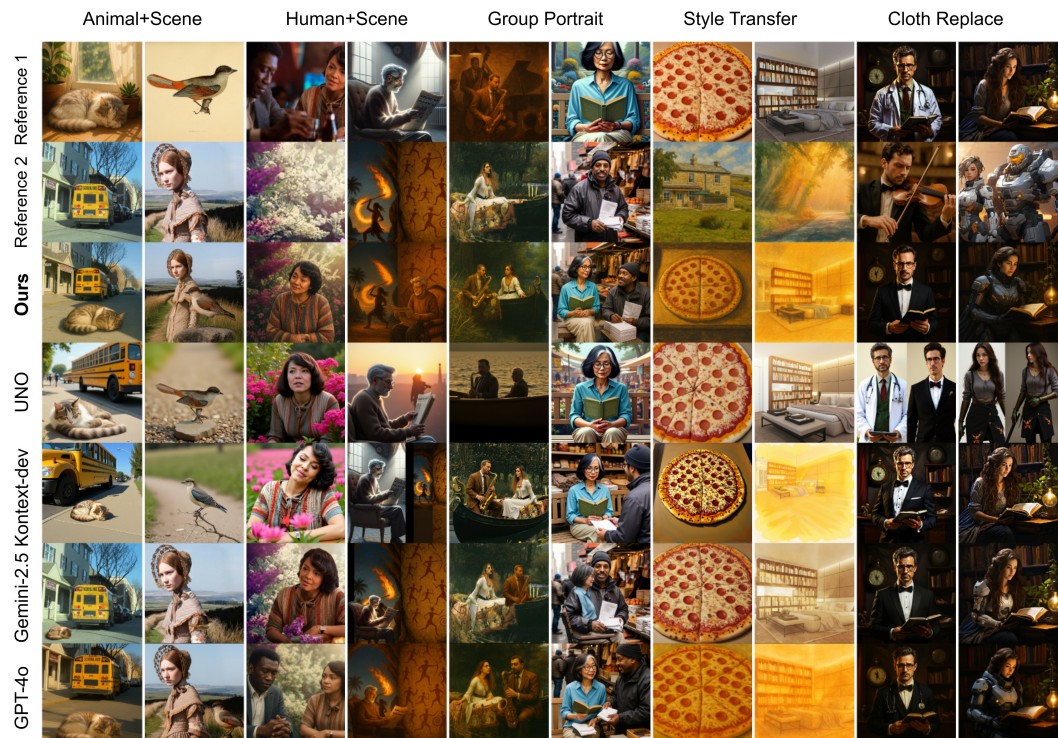

Figure 2: Illustration of the model framework. The pretrained model for single-image editing is modified in two ways: (1) a diagonal concatenation strategy for image position embeddings, and (2) LoRA adaptation for finetuning without degrading single-image editing.

given background with consistent perspective. Third, *group portraits* focus on merging two people into a coherent group image. In *style transfer*, the visual style of one reference image is applied to the content of another. Finally, *clothing replacement* involves transferring clothing appearance from one subject to another.

**Data generation**. Input image pairs are sampled from datasets built in HQ-Edit, OmniEdit, and UltraEdit, where images are grouped according to semantic themes such as animals, scenes, or clothing. For each sampled pair, GPT is prompted to produce (i) natural-language editing instructions and (ii) a structured editing plan specifying semantic constraints (e.g., object placement, cross-image consistency, identity preservation). This process yields high-quality instruction–input–output triples that support supervised training for dual-image editing.

This dataset serves two roles: it trains our model and acts as the first benchmark for evaluating dual-image editing performance.

## 3.2 EXTENDING MODEL FOR DUAL-IMAGE FUSION

The Flux Kontext-dev model is a strong open-source editor but is designed for single-image positional embeddings. Directly inputting multiple references leads to positional conflicts and degraded performance.

**Positional embedding modification**. We applied the dual positional embedding fusion mechanism as illustrated in Figure 2. Each reference image is encoded separately through the standard Flux encoder. Positional embeddings are projected into a shared latent space and fused using a gated cross-attention mechanism. A diagonal concatenation strategy is applied during fusion, which not only preserves spatial correspondence from both inputs but also allows the model to handle reference images of mismatched aspect ratios without distortion. The fused embeddings remain fully compatible with pretrained weights, ensuring stable training.

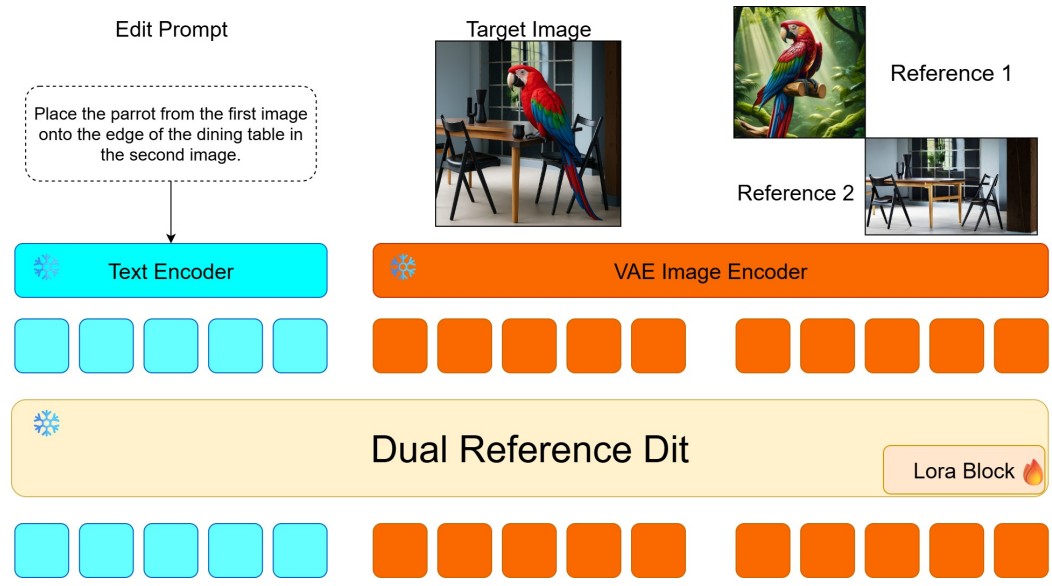

Figure 3: Reinforcement learning pipeline. Input edit pairs are sampled to generate edits via inference. The edited results and inputs are combined with prompts for VLM evaluation. The evaluation scores are used to compute advantages, which update the model parameters for the next batch

**LoRA fine-tuning**. To efficiently adapt the model without retraining from scratch, we employ Low-Rank Adaptation (LoRA)(Hu et al., 2022). This lightweight fine-tuning enables the Flux backbone to acquire dual-image capabilities while avoiding catastrophic forgetting of single-image editing.

### 3.3 REINFORCEMENT ALIGNMENT WITH BINARY VLM FEEDBACK (DDPO)

Supervised training grants initial dual-image capability, but generations can drift from user intent. We therefore perform a second-stage alignment using Diffusion Denoising Policy Optimization (DDPO), driven by a binary vision–language judge (GPT-5 VLM) for stability. Our pipeline in shown in Figure 3.

**Rollouts on the Supervised Data**. Given the supervised training tuples $(x_1, x_2, t)$ (two reference images and an instruction), we sample an edited image $\hat{y} \sim p_\theta(\cdot \mid x_1, x_2, t)$ using the current diffusion policy $p_\theta$. To reduce overfitting and reward hacking, we (i) randomize seeds and guidance scales, and (ii) use small ensembles of sampling parameters during rollout.

**Binary VLM Judge (Yes/No Only).** For each $(x_1, x_2, t, \hat{y})$, the VLM is prompted with three independent yes/no questions that assess different aspects of generation quality. The first evaluates *instruction fidelity*, asking whether $\hat{y}$ follows the textual instruction $t$. The second checks *dual-reference consistency*, determining whether visual attributes from both $x_1$ and $x_2$ are faithfully integrated. The third concerns *visual realism*, judging whether $\hat{y}$ is coherent and artifact-free. The VLM is constrained to output "yes" or "no" (with no scores but explanations), as binary judgments have proven more stable than open-ended scoring and are less sensitive to prompt drift. We denote the resulting binary indicators as $b_{\text{fid}}, b_{\text{cons}}, b_{\text{real}} \in \{0, 1\}$, where 1 corresponds to "yes" and 0 to "no."

**Reward and Advantage**. We compute a scalar reward by averaging the three binary indicators:

$$r = \frac{1}{3}(b_{\text{fid}} + b_{\text{cons}} + b_{\text{real}})$$

We estimate an advantage $A = r - b$ using a moving-average baseline $b$. This yields low-variance, well-calibrated credit assignment without learning a separate value function.

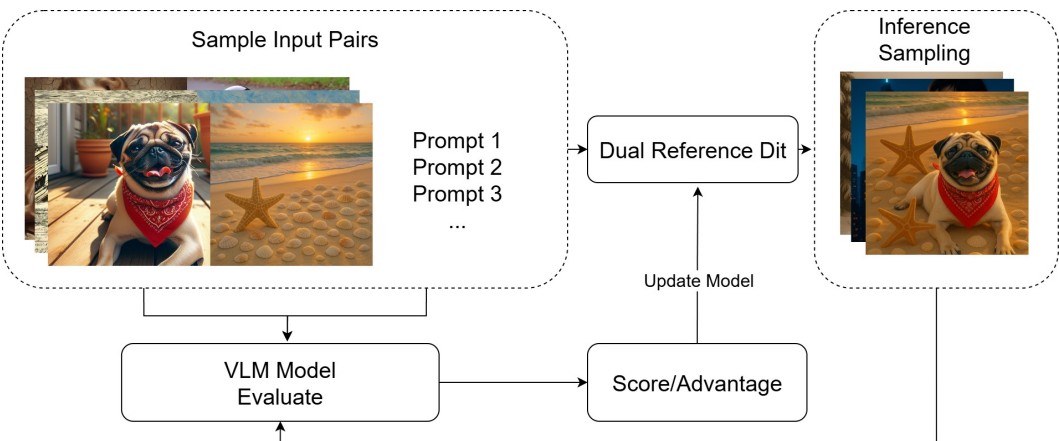

Figure 4: Qualitative comparison. GPT-4o-Image achieves the best editing quality. Our dual-reference model follows instructions well but preserves identity less consistently. UNO handles only limited tasks, such as object merging. Flux Kontext, our base model, is easily misled by concatenated inputs. Gemini 2.5 shows comparable quality with stronger identity preservation.

**DDPO Update**. We follow DDPO to update the diffusion policy parameters $\theta$ across the denoising trajectory:

$$\nabla_\theta \mathcal{J}(\theta) \approx \mathbb{E}_{\hat{y} \sim p_\theta} \left[ A \sum_{s=1}^{S} \nabla_\theta \log p_\theta(\varepsilon_s \mid z_s, x_1, x_2, t) \right].$$

where $s$ indexes denoising steps, $z_s$ are latent states, and $\varepsilon_s$ are noise predictions. Practically, we weight per-step losses by $A$ and apply PPO-style clipping (DDPO) to stabilize large updates. We keep LoRA adapters trainable and freeze the bulk of the backbone, preserving single-image skills.

## 4 EXPERIMENTS

### 4.1 EXPERIMENTAL SETTING

We build our experiments on the Flux Kontext-dev (open-source release), adapting it for dual-image editing. Training is conducted on our synthetic dataset (Sec. 3.1), using LoRA fine-tuning with a rank of 512 and a learning rate of $5 \times 10^{-5}$. Reinforcement alignment with DDPO is applied after supervised training with lora-rank 32 and a learning rate $5 \times 10^{-5}$.
All experiments are conducted on a server with 8 × H100 80GB GPUs. Unless otherwise specified, outputs are generated at 512×512 resolution with 28 sampling steps.

### 4.2 BENCHMARK WITH SYNTHETIC DATA

We evaluate our method and baselines on the five dual-image tasks introduced in Sec. 3.1: (1) animal–scene composition, (2) person–scene insertion, (3) group portraits, (4) style transfer, and (5) clothing replacement. Each task contains 100 test pairs of reference images, forming a balanced benchmark. A key challenge is that standard metrics such as CLIP-SIM or DINO similarity fail to capture fusion quality across two references and often penalize valid stylistic deviations. Therefore, we adopt a vision–language model (VLM)-based evaluation, using GPT-5 and SeedVLM as evaluation models. For each test case, VLM receives the instruction, the two reference images, and the generated output, and returns binary judgments on instruction fidelity, dual-reference consistency, and visual realism. The three indicators are averaged into a scalar score, and the final evaluation metric is the mean score over all 500 samples. Additionally, to assess whether single-reference editing is preserved after dual-reference finetuning, we benchmark the finetuned model against the base model on GEdit-Bench(Liu et al., 2025).

Table 1: Benchmark with GEdit-Bench for single-ref edit.

| Model | Score(GPT-5) |
|---|---|
| Native Flux.1-Kontext-dev | 5.9106 |
| Dual-image finetuned(ours) | 6.2076 |

Table 2: Comprehensive evaluation results across different benchmarks.

| Model(GPT-5) | Animal | Person | Group | Style | Clothing |
|---|---|---|---|---|---|
| UNO | 0.600 | 0.620 | 0.533 | 0.636 | 0.530 |
| Flux-kontext-dev | 0.720 | 0.667 | 0.663 | 0.773 | 0.480 |
| Ours | 0.806 | 0.840 | *0.832* | 0.929 | 0.741 |
| Ours (RL) | *0.843* | *0.867* | 0.820 | *0.950* | 0.757 |
| Gemini | 0.793 | 0.746 | 0.807 | 0.817 | *0.870* |
| GPT | **0.933** | **0.947** | **0.933** | **0.977** | **0.950** |

| Model(SeedVLM) | Animal | Person | Group | Style | Clothing |
|---|---|---|---|---|---|
| UNO | 0.766 | 0.656 | 0.563 | 0.650 | 0.536 |
| Flux-kontext-dev | 0.820 | 0.773 | 0.747 | 0.813 | 0.573 |
| Ours | 0.870 | 0.873 | 0.860 | 0.957 | 0.860 |
| Ours (RL) | *0.880* | *0.880* | *0.903* | *0.963* | *0.873* |
| Gemini | 0.710 | 0.727 | 0.833 | 0.807 | 0.817 |
| GPT | *0.960* | *0.970* | *0.990* | *0.997* | *0.937* |

**Models for Comparison**. We compare our approach with both closed- and open-source systems:

- **Closed-source:** GPT-4o and Gemini-Image-Flash 2.5 (Nano Banana).
- **Open-source:** UNO (Less-to-More Generalization), and Flux Kontext-dev. Since Flux does not natively support multi-image inputs, we concatenate the two reference images as a baseline.
- **Ours:** Flux + Dual Positional Embedding (LoRA), and Flux + Dual Positional Embedding + RL (DDPO).

## 4.3 EXPERIMENTAL RESULTS

**Single-image performance**. Table 1 and Figure A.1 shows that our dual-image extension maintains editing quality on single-image instructions, matching the performance of the original Flux Kontext. Importantly, the extension does not degrade single-image ability; instead, after supervised training on our synthetic tasks, the model achieves performance comparable to Gemini-Image-Flash 2.5, though still behind GPT-4o. Compared to the vanilla Flux, the improvements are substantial, confirming that our positional embedding modification is lossless.

**Dual-image performance**. Table 2 and Figure 4 demonstrates that our method effectively understands and fuses multiple references, achieving the best results among open-source systems and setting a new state of the art. Reinforcement alignment with DDPO further improves instruction fidelity and reference consistency, yielding an additional $\approx 3\%$ gain over supervised training alone. This enhancement allows our full model to surpass Gemini in dual-image tasks, particularly in person–scene insertion and group portraits, where dual reference integration is critical.

### 4.4 ABLATION STUDY

We further compare models with and without DDPO in figure 5. Without RL, outputs sometimes deviate from instructions (e.g., incomplete object insertion, style mismatch). RL alignment increases instruction fidelity and overall GPT-5 reward by average of 3%, showing the benefit of reinforcement learning in dual-image editing.

## 5 CONCLUSION

We presented the first open-source, high-quality dual-image reference dataset together with a standardized benchmark for multi-image editing. Building on this, we introduced a baseline method that extends the widely used Flux Kontext model with a lossless dual-reference embedding mechanism, enabling robust multi-image editing without sacrificing single-image ability. Finally, we released an open RL-based alignment framework that allows further fine-grained tuning using vision–language feedback. We hope these contributions provide a solid foundation for the community to advance research in instruction-driven multi-image editing.

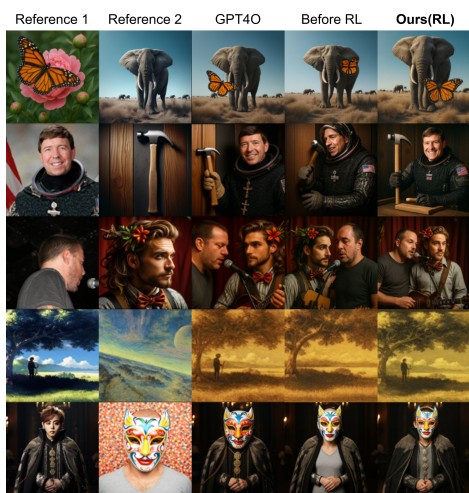

Figure 5: Visualization of reinforcement learning. RL improves both identity preservation and instruction following.

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

## A  APPENDIX

Large Language Models was used to polish the writing of this paper.

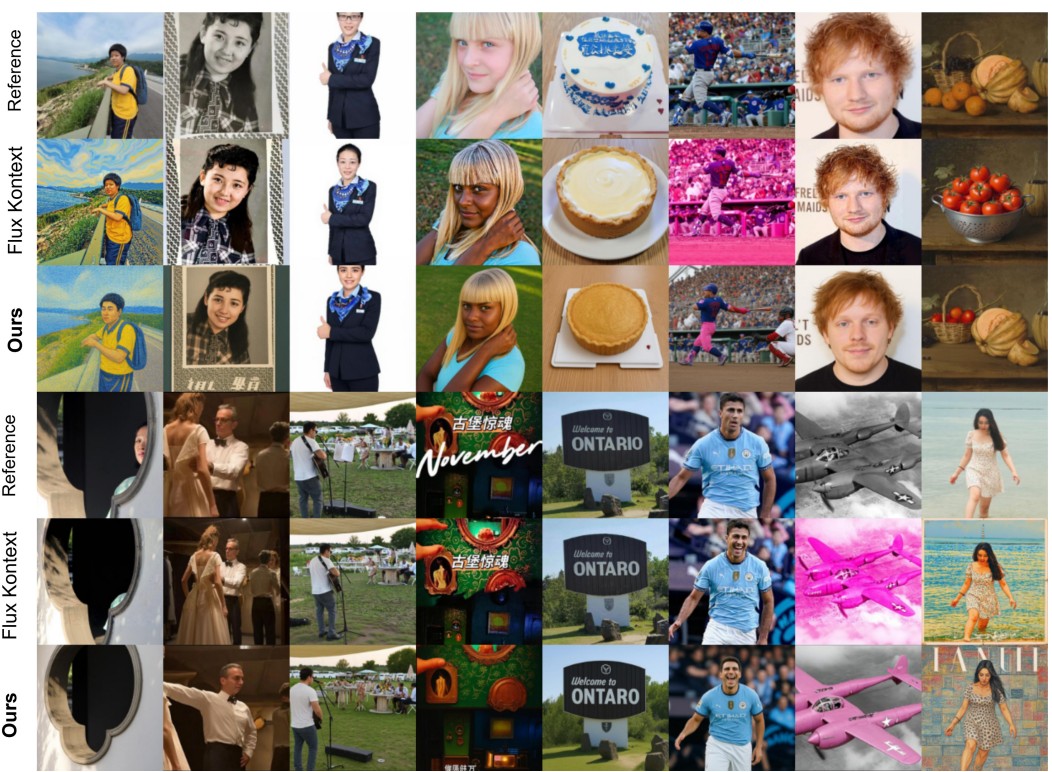

Figure A.1: Single-image reference editing. Benchmark of finetuned dual-image reference model against the original Flux Kontext-dev baseline on GEdit-Bench. Both models achieve comparable editing quality and instruction-following performance, and the ability to handle single-reference edits is preserved after finetuning.

