# OpenReview forum: "From Single to Dual Reference: Reinforcement-Aligned Multi-Image Instruction-Guided Editing"
_ICLR.cc/2026/Conference — Submitted to ICLR 2026_

### Official Review · Reviewer_rxTc · 2025-10-28

**Soundness:** 2
**Presentation:** 2
**Contribution:** 2
**Rating:** 2
**Confidence:** 3

**Summary:**

This work presents a dual-image instruction-guided editing, addressing a significant gap in current open-source image editing models, which primarily focus on single-image references. To facilitate this, a novel synthetic dataset comprising five representative dual-image editing tasks (e.g., animal-scene composition, group portraits) is constructed. The proposed method extends the Flux Kontext-dev model by incorporating a dual positional embedding scheme with LoRA fine-tuning, enabling efficient multi-reference fusion without catastrophic forgetting of single-image capabilities. Furthermore, reinforcement alignment utilizing Diffusion Denoising Policy Optimization (DDPO) with a Vision-Language Model (VLM) (GPT-5) as a reward model is employed to enhance instruction-following and consistency. The approach demonstrates qualitative and quantitative experiments with baselines to assess its performance against existing methods.

**Strengths:**

This work proposes an architectural extension for an existing single-image editor, utilizing a dual positional embedding scheme and LoRA fine-tuning for multi-reference fusion without compromising prior capabilities. Additionally, reinforcement alignment with a VLM reward model is integrated to enhance instruction following. The reported results indicate performance surpassing several open-source baselines and achieving comparable or superior outcomes to certain closed-source models in specific tasks.

**Weaknesses:**

1. Lack of Novelty in the Proposed Method. The proposed method lacks novelty, as it simply applies LoRA fine-tuning to the Flux model using DDPO in a rather naive manner. The approach does not present any substantial originality or methodological innovation beyond existing techniques.

2. Sole reliance on VLM-based evaluation is insufficient for a comprehensive assessment. The absence of objective metrics, such as CLIP score or CLIP directional similarity, limits a robust

3. The qualitative results also fail to demonstrate superior performance compared to other models. As shown in Figure 2, Gemini-2.5 and GPT-4o preserve the originality of the images more effectively while better incorporating the provided references than the proposed method. This raises concerns about the robustness and reliability of the proposed approach.

4. A demo link provided in the abstract of this paper reveals author-identifying information, specifically an account associated with the authors’ names. This raises a serious concern of violating ICLR’s double-blind review policy, as the information could potentially compromise the authors’ anonymity.

**Questions:**

Please see the weakness

---

### Official Review · Reviewer_HxYW · 2025-10-28

**Soundness:** 3
**Presentation:** 3
**Contribution:** 2
**Rating:** 2
**Confidence:** 3

**Summary:**

Paper: From Single to Dual Reference: Reinforcement-Aligned Multi-Image Instruction-Guided Editing.
The paper tackles dual-image instruction-guided editing (e.g., person–scene insertion, clothing transfer). It (i) builds a synthetic, task-balanced benchmark of five categories and 100 test pairs each; (ii) extends a single-reference editor (Flux Kontext-dev) with a dual positional-embedding fusion and LoRA fine-tuning to avoid catastrophic forgetting (model diagram on p.4, Fig. 2); and (iii) applies DDPO with a binary VLM judge (three yes/no questions for instruction fidelity, dual-reference consistency, and realism) to align generations (p.5, Fig. 3). Quantitatively, the method outperforms open baselines (UNO, Flux concat) and approaches or surpasses Gemini on several dual-image tasks; RL adds ~3% average reward over supervised fine-tuning (p.7, Table 2; p.8, Fig. 5). Single-image editing ability is preserved (p.7, Table 1; p.10, Fig. A.1).

**Strengths:**

Problem focus & gap: Dual-image instruction editing is practical yet underexplored; the paper formulates clear tasks and success criteria (p.2–3).
Architectural clarity: The dual positional-embedding fusion (diagonal concatenation + cross-attn) is compatible with pretrained weights and handles aspect-ratio mismatch (p.4, Fig. 2).
Practical alignment: Binary-question VLM rewards reduce prompt drift and stabilize DDPO; the 3-question design is sensible (p.5–6).
Empirical signal: Gains over open baselines across all five tasks on two evaluators; RL adds ~3% reward on average (p.7–8, Table 2 & Fig. 5).
No regression on single-image: Single-ref performance is retained after dual-ref finetuning (p.7, Table 1; p.10, Fig. A.1).

**Weaknesses:**

Evaluator dependence / generality. The main metric is VLM-based (GPT-5, SeedVLM). No human study or reference-free classical metrics (e.g., FID-like realism proxies, identity/text checks) are reported; this risks evaluator bias.
Synthetic benchmark only. The testbed is entirely synthetic (500 total samples). Lack of real-image or hard OOD cases (e.g., clutter, strong occlusion, lighting mismatch) limits external validity.
Ablation depth. Beyond showing RL vs. no-RL, there is little analysis of the fusion design (e.g., alternatives to diagonal concat, gating choices), LoRA ranks/targets, or reward shaping (weights per question).
Identity preservation quantification. The qualitative grid (p.6, Fig. 4) notes weaker identity retention than GPT-4o; no ID metric (face embedding similarity, retrieval) is reported for the human categories.
Reproducibility details. Training uses 8×H100 with certain step counts; more explicit compute budgets, wall-clock, and exact prompts/thresholds for the judge would help.

**Questions:**

Human evaluation: Can you add a small human study (e.g., 100 cases, majority vote) to calibrate VLM scores and report agreement vs. GPT-5/SeedVLM?
Fusion ablations: How does performance change with alternative fusions (concat-then-attn; interleaved tokens; learned offset pe) and different LoRA ranks/targets? A table analogous to Table 2 would strengthen causality.
Reward shaping: You average three binary questions (fid/cons/real). Did you try weighted rewards or min-operator (pessimistic) to better penalize failures? Any reward-hacking symptoms?
Identity metrics: For person/group tasks, can you report ID-similarity (ArcFace/FaceNet) and pose/landmark consistency to substantiate claims in Fig. 4?
Beyond two references: Is the fusion scalable to >2 references (e.g., three-person composites)? Any preliminary scaling results or constraints?
Real-world/OOD: Can you include a small real-image slice (non-synthetic) and a hard-case subset (strong occlusion, lighting shifts) to test robustness?
Additional Feedback (constructive)

Add Pareto plots (instruction fidelity vs. reference consistency; realism as contours) to visualize trade-offs across models/ablations.
Report CIs/SEs for all means in Table 2; include per-category difficulty analysis and typical failure taxonomies.
Release judge prompts and decision heuristics (box sizes, thresholds) in an appendix table; provide a seed list for rollouts.
Consider a compact open-stack evaluator (e.g., GLIP/OWL-ViT + open VLM) for community replication alongside GPT-5/SeedVLM numbers.

**Details Of Ethics Concerns:**

The paper uses synthetic data and focuses on evaluation/training methodology. If any human-subject images are used in future releases, include license/consent and identity-preservation caveats.

---

### Official Review · Reviewer_13cT · 2025-10-31

**Soundness:** 3
**Presentation:** 3
**Contribution:** 2
**Rating:** 4
**Confidence:** 4

**Summary:**

This paper systematically investigates dual-image instruction-guided image editing, which is a task that has not been sufficiently explored. Most existing models only support single-image inputs, making them inadequate for real-world scenarios that require fusing semantic information from two reference images.

**Strengths:**

1. The proposed dual-image editing benchmark fills a critical gap in the field by providing the first standardized dataset for multi-reference image editing.
2. The experiments are thorough, evaluating performance not only on the newly introduced benchmark but also verifying the preservation of single-image editing capabilities.
3. The paper is clearly written, and the network architecture diagram effectively illustrates the model workflow.

**Weaknesses:**

Major Weaknesses
1. Limited data quality: All training and test data are synthesized by combining GPT-generated instructions with publicly available images. The resulting image pairs may lack physical plausibility (e.g., in lighting, scale, or occlusion), potentially causing the model to learn spurious correlations.
2. Limited methodological novelty: Prior work, such as KOSMOS-G (arXiv:2310.02992), has already explored integrating multimodal large language models (MLLMs) with diffusion models to handle complex prompts involving multiple images and interleaved text, achieving similar functionality. Although the technical approaches differ, the core challenge—“instruction-guided multi-image editing”—is notably overlapping, which significantly diminishes the perceived novelty of this work.

Minior Weaknesses
1. Unverified evaluation capability: The paper uses GPT-5 as the reward model for DDPO, but it remains unclear whether GPT-5 actually has the demonstrated ability to judge visual fidelity, semantic alignment, or multi-image consistency in image editing tasks.

**Questions:**

See Weaknesses

---

### Official Review · Reviewer_67Z7 · 2025-11-01

**Soundness:** 2
**Presentation:** 1
**Contribution:** 2
**Rating:** 2
**Confidence:** 3

**Summary:**

This paper introduces a dual-reference image editing framework that extends single-image instruction-guided editing models (e.g., Flux Kontext) to handle two input images. The authors design a synthetic dataset covering five dual-image tasks and propose a dual positional embedding mechanism with LoRA fine-tuning to enable efficient fusion. Additionally, reinforcement alignment using DDPO with a vision–language model (GPT-5) is applied to improve instruction adherence. Experiments show improvements over existing open-source baselines and competitive results compared to proprietary systems like Gemini.

**Strengths:**

1. The paper presents the first systematic benchmark and pipeline for dual-reference image editing, addressing a meaningful technical gap in the current literature.

2. The use of DDPO for vision–language feedback is technically sound and shows potential to enhance model alignment with instructions.

**Weaknesses:**

1. Unclear motivation： The motivation is not well articulated. From the examples shown, it remains unclear whether the proposed setting truly requires textual instructions; yet the title and framework emphasize instruction-guided editing. The paper should clarify the real necessity of instructions in dual-reference fusion tasks.

2. Poor writing and layout quality: The manuscript appears rushed. Starting from Figure 2, several figures and captions are mismatched or misplaced, which severely affects readability. The paper needs substantial revision in both writing and formatting before it can meet publication standards.

**Questions:**

Please see the weaknesses.

---

### Meta-Review · Area_Chair_QuJV · 2026-01-05

**Summary:**

The paper receives review scores of 2224 with unanimous rejections (limited novelty by Reviewer 13cT and Reviewer rxTc, validity of evaluation metric by HxYW and rxTc, unclear motivation & poor writing by Reviewer 67Z7, anonymity issue with an html link including potential author info raised by rxTc). Although there are many issues and reviews are very negative, there is no rebuttal from authors. There is also a concern raised on breaking anonymity issues with hyperlink in the paper.

**Reviewer Concerns:**

Anonymity issues with the hyperlink is an outstanding and serious issue.
No any concerns addressed as there is no rebuttal.

**Reviewer Scores:**

All reviewers will keep reject as there is no rebuttal.

---

### Decision · Program_Chairs · 2026-01-26

Reject